# A New Research Tool for Use in Sharks and Rays: Relevance of Reproductive Hormone Levels in the Skin of Small-Spotted Catshark (*Scyliorhinus canicula*)

**DOI:** 10.3390/ani15050762

**Published:** 2025-03-06

**Authors:** Annaïs Carbajal, Isabel González Lobato, Clara Agustí, Marta Muñoz-Baquero, Paula Serres-Corral, Manel López-Béjar

**Affiliations:** 1Department of Animal Health and Anatomy, Veterinary Faculty, Universitat Autònoma de Barcelona, 08193 Bellaterra, Spain; paula.serres@uab.cat (P.S.-C.); manel.lopez.bejar@uab.cat (M.L.-B.); 2Associació Aletea, 17488 Cadaqués, Spain; isalobato94@gmail.com (I.G.L.); c.agustipujol@gmail.com (C.A.); 3Animal Welfare Education Centre (AWEC), Veterinary Faculty, Universitat Autònoma de Barcelona, 08193 Bellaterra, Spain; 4Fundació Oceanogràfic de la Comunitat Valenciana, Research Department, Ciudad de las Artes y las Ciencias, 46005 Valencia, Spain; mmunoz@oceanografic.org; 5Departamento de Producción y Sanidad Animal, Salud Pública Veterinaria y Ciencia y Tecnología de los Alimentos, Instituto de Ciencias Biomédicas, Facultad de Veterinaria, Universidad Cardenal Herrera-CEU, CEU Universities, 46115 Alfara del Patriarca, Spain

**Keywords:** non-invasive sample, reproduction, testosterone, progesterone, 17β-estradiol, chondrichthyans, conservation

## Abstract

Given the conservation concerns surrounding many chondrichthyan species and the growing importance of reducing the use of animals in research and minimizing their suffering, we should prioritize the use of methods that avoid harming or killing the animals. Although blood is the “gold standard”, applying standard blood collection protocols is not always feasible for chondrichthyans living in the wild, especially because these methods require chase, capture, and restraint. The present study was performed to evaluate whether sex steroid hormones detected in shark skin, a sample that can be collected remotely, provide relevant information about their reproductive state. The analytical methodology was first validated, demonstrating that the method can reliably quantify sex steroid hormones in shark skin. After ensuring that reproductive hormones are accurately measured in this sample type, we compared hormone levels between males (mature and immature) and females. Although further validation is required, differences detected between sexes suggest that skin may be a promising alternative approach for obtaining reproductive data in free-ranging sharks.

## 1. Introduction

Despite efforts to avoid chondrichthyan bycatch or unintentional catch by large fisheries, many populations remain threatened or endangered, and many of them are subject to a myriad of other anthropogenic impacts [1,2,3]. According to the IUCN Red List, over one-third of chondrichthyan species are estimated to be threatened [4,5]. Gathering data about all aspects of their biology, behavior, and life history is relevant for the proper conservation and management of shark and ray populations. Nevertheless, obtaining such information without handling the animal or using lethal methods can be extremely difficult. 

Traditional physiological analyses of plasma samples have, to date, provided essential information on the basic endocrine system of chondrichthyans [6,7]. However, obtaining blood samples is not always possible or desirable, especially in field conditions. Blood drawing may require physical or chemical restraint, which may in turn affect the biology of the specimen and impact the target endocrine measurements [8,9]. Considering the threatened status of many chondrichthyan species [1,5], as well as the need to enforce the implementation of the 3R principles in wildlife research [10], efforts should be focused on obtaining in-depth information using non-lethal or minimally invasive samples collected remotely. Endocrinological studies using alternative samples can offer a potential avenue forward. In other taxa, less invasive samples, some of which can be collected without the need to chase, capture, or restrain the animal, are being frequently applied by scientists without major disruption to the animal [11,12]. These sample types, such as feces, hair, saliva, feathers, or blubber, are more easily collected than blood and have proven to provide reliable physiological data [13].

In sharks, diverse measures on skin biopsies have been applied to understand genetics, the impact of contamination, and dietary information, among others [14,15,16]. This sample type can be remotely collected using darts, thus avoiding capture, killing, or relying on dead animals for tissue sampling. Biopsies are usually taken using a pole attached to a remote biopsy sampling tip that can penetrate the thick skin of sharks [14,17]. Recently, a study demonstrated for the first time that chondrichthyans’ skin contains measurable levels of testosterone [18]. While that study validated essential methodological aspects regarding the analysis of testosterone in shark skin, it is crucial to verify whether hormone levels in this sample type provide meaningful physiological data [19,20]. The structural and physiological similarities between shark skin and tissues known to accumulate steroid hormones provide a compelling rationale for investigating the presence of these hormones in shark skin. Like teeth, hair, and fish scales, shark skin comprises a complex interplay of cellular layers and an extracellular matrix, with close connections to the circulatory system. Dermal denticles, homologous to vertebrate teeth, originate in the dermis and project through the epidermis, potentially providing a matrix for hormone binding. The dermis itself is highly vascularized [21,22], allowing for the efficient transport of blood-borne substances, including reproductive hormones, from capillaries into the denticle cells and surrounding skin tissues.

Accordingly, the present study aimed to (i) develop a suitable extraction and analytical technique for reproductive hormone determination in the skin of small-spotted catshark (*Scyliorhinus canicula*, Linnaeus, 1758) and (ii) compare hormone levels between males (mature and immature) and females to evaluate the variation in reproductive hormones relative to sex and males’ maturity stage. Progesterone, 17β-estradiol, and testosterone were selected given the strong relationship between these three hormones and the reproductive activity in several chondrichthyan species [7,23,24]. In addition, our current understanding of how hormones regulate reproduction in sharks is mainly based on studies on these three hormones, given that they are the main sex steroids in vertebrates [25,26].

The small-spotted catshark was selected for this study due to its suitability as a model species. Its relatively small size, ease of handling, and accessibility in aquaria and local fish markets facilitate research. While globally abundant [27], *S. canicula* exhibits sensitivity to overexploitation and pollution, highlighting the need for further investigation [28,29,30]. Therefore, understanding the reproductive biology of this species is crucial for effective conservation management. Measurements of reproductive hormones have been used to evaluate individual reproductive status, providing relevant information regarding sex, maturity stage, and the reproductive cycle of the animal [25,31]. Knowledge of the reproductive status of a certain population can, therefore, inform understanding of demographics and viability, both of which are crucial to the development of science-based conservation and management plans [19,32].

## 2. Materials and Methods

### 2.1. Animals and Sampling

Skin samples were collected from 41 (21 females and 20 males) necropsied small-spotted catshark (*Scyliorhinus canicula*). Individuals were donated from local fisheries from Valencia (Spain, 39°26′45″ N 0°19′12″ W) as they had been accidentally captured as part of commercial and artisanal activities. Immediately after being captured, specimens were frozen at −20 °C. Afterwards, individuals were sent frozen to the Veterinary Faculty of the Universitat Autònoma de Barcelona, where the hormonal extraction and analyses were performed. 

Once in the laboratory, sex, total length, and body weight were recorded (Table 1). Fulton’s condition factor, used to reflect an individual’s energetic state [33], was calculated following the formula K = 106 × body weight (g) × total length (mm) − 3 [34]. The maturity of males was recorded based on the biometric information obtained and the presence of sperm in their reproductive tract. After identifying the maturity stage, the gonads’ weight was registered and used to measure the gonadosomatic index (GSI; gonad weight expressed as a percentage of body weight). The reproductive stage could not be evaluated in the group of females as their reproductive system had been previously removed and the data were not recorded.

Skin samples with a mean size of 2.77 × 2.37 cm (length × width) were collected caudal to the first dorsal fin using a scalpel and tweezers. First, the region was sectioned with the scalpel, and then the skin was separated from the body with tweezers. If muscle remnant was still present in the sample, it was completely removed with the scalpel. 

### 2.2. Hormone Extraction

The hormone extraction was performed following a previously validated protocol for small-spotted catshark [18]. Briefly, skin samples were first cleaned with isopropanol to remove any external contamination. Then, samples were dried for 24 h in an oven at 60 °C. Once completely dry, skin samples were mechanically milled in particles of <1 mm with a ball mill. Milled samples (50 mg) were mixed with 1.5 mL of pure methanol and incubated at 30 °C overnight for hormone extraction. Samples were then centrifuged, and the supernatant was placed in an oven until its complete evaporation. After 24 h, extracts were reconstituted with EIA buffer solution provided by the EIA kit (Neogen^®^ Corporation, Ayr, UK) and immediately stored frozen until hormone determination.

### 2.3. Hormone Analysis and Assay Validation

Commercial enzyme immunoassays of testosterone, estradiol, and progesterone (EIA KIT; Neogen^®^ Corporation, Ayr, UK) were used to quantify hormone levels in skin extracts of small-spotted catshark. According to the manufacturer, cross-reactivity of the antibody with other steroids is as follows: for testosterone measurements, testosterone 100%, dihydrotestosterone 100%, androstenedione 0.86%, bolandiol 0.86%, testosterone enanthate 0.13%, estriol 0.10%, and testosterone benzoate 0.10%; for 17β-estradiol measurements, 17β-estradiol 100%, testosterone 1.0%, estriol 0.41%, and estrone 0.10%; and for progesterone measurements, progesterone 100%, deoxycorticosterone 2.5%, corticosterone 2.0%, pregnenolone 2.0%, androstenedione 1.0%, 17-hydroxyprogesterone 0.4%, testosterone 0.29%, cortisol 0.2%, cortisone 0.2%, dehydroepiandrosterone 0.2%, estradiol 0.2%, estriol 0.2%, and estrone 0.2%. Steroids with cross-reactivity < 0.06% are not presented.

Skin extracts from 10 different males and 10 different females were pooled separately for assay validation. Precision was calculated with the intra-assay coefficient of variation (CV) given by all duplicated samples measured. The specificity of the assay was tested with the linearity of the dilution, determined by using dilutions of the pools mixed with the EIA buffer. Accuracy was evaluated with the spike-and-recovery test, measured by adding different volumes of the pool (25, 75, and 100 µL) to known volumes (25, 75, and 100 µL) and concentrations of pure standard (testosterone, 0.008 ng/mL, 0.02 ng/mL, and 0.04 ng/mL; 17β-estradiol, 0.2 ng/mL, 0.4 ng/mL, and 1 ng/mL; and progesterone, 2 ng/mL, 4 ng/mL, and 10 ng/mL) solution. Finally, the sensitivity of the test was given by the smallest amount of hormone that the assay was able to distinguish and accurately measure from the skin extracts. 

### 2.4. Statistical Analysis

Data obtained were analyzed using R software (R-project, Version 4.2.2, R Development Core Team, University of Auckland, Auckland, New Zealand) with a *p*-value below 0.05 as a criterion for significance. First, the assumption of normality was checked using a Shapiro–Wilk test, and concentrations were log transformed where necessary to achieve normality. When data were not normally distributed, non-parametric tests were applied.

For assay validation, the relationship between expected and obtained values from the dilution test were determined using Pearson’s Product correlation test. Accuracy results were assessed as the total concentration expected (standard + skin pool) compared to obtained levels and evaluated using Pearson’s correlation test.

To evaluate whether the skin of males (immature and mature separately) and females differed in sex hormone concentrations, one-way ANOVAs with Tukey’s pairwise post hoc tests were applied. To further study the relationship between males’ maturity and sexual hormone concentrations, Pearson correlation tests between skin testosterone levels and male GSI were performed (with mature and immature specimens tested separately). The same tests were applied to study the relationship between Fulton’s condition and the hormonal levels.

## 3. Results

The results of assay validation are shown in Table 2. All reproductive hormone concentrations measured in shark skin samples were above the detection limits of their respective assays. As demonstrated by the Pearson’s Product correlation, serial dilutions of skin extracts displayed significant correlation coefficients between obtained and expected values (for all three hormones studied and for both sexes; r = 0.99, *p* < 0.01). Similarly, in the spike-and-recovery test, expected values were significantly correlated to obtained concentrations (for all three hormones studied and for both sexes; r > 0.80, *p* < 0.01). 

The results of ANOVA (Figure 1A; F_2,38_ = 10.58, *p*-value = 0.01) revealed significant variations in skin progesterone concentrations. The subsequent Tukey’s post hoc analysis showed that females exhibited significantly higher concentrations (73.0 ± 28.5 pg/mg) than both immature (38.4 ± 21.5 pg/mg; *p*-value = 0.01) and mature (37.5 ± 22.6 pg/mg; *p*-value < 0.001) males. Similarly, differences in skin 17β-estradiol levels were detected (ANOVA, Figure 1B; F_2,38_ = 12.6, *p*-value < 0.001). Females presented significantly higher 17β-estradiol levels (4.3 ± 1.2 pg/mg) in comparison to immature (2.5 ± 0.9 pg/mg; *p*-value < 0.01) and mature (2.6 ± 1.1 pg/mg; *p*-value < 0.001) males. No differences were detected between immature and mature males in terms of progesterone (Tukey’s test, *p*-value > 0.05) or 17β-estradiol (Tukey’s test, *p*-value > 0.05) concentrations. Regarding skin testosterone (ANOVA, Figure 1C; F_2,38_ = 4.81, *p*-value = 0.01), the post hoc test revealed that while mature males (2.7 ± 1.0 pg/mg) had significantly higher hormone concentrations than females (2.1 ± 0.3 pg/mg; *p*-value = 0.014), no differences were present between immature (2.1 ± 0.5 pg/mg) and mature males (*p*-value = 0.11) or between immature males and females (*p*-value = 0.99). 

An analysis of the relationships between the three hormones studied and the Fulton condition in mature males, immature males, and females revealed no significant correlations (Figure 2; *p* > 0.05). However, a notable exception emerged in immature males, where a positive correlation was observed between skin testosterone levels and the Fulton condition (r = 0.83, *p*-value = 0.04). No significant correlation between testosterone concentrations in males and GSI was detected (mature, r = −0.15, *p*-value = 0.60; immature; r = −0.09, *p*-value = 0.87).

## 4. Discussion

This study represents the first major step in validating the use of skin biopsies for measuring reproductive steroid hormone concentrations in a shark species, the small-spotted catshark, and evaluating their variation relative to sex and males’ maturity stage. 

We detected significant differences in skin hormone levels between sexes, suggesting that this technique may be a promising alternative approach for obtaining reproductive data in free-ranging sharks. The commercially available EIA kits validated in the present study were sensitive enough to measure progesterone, 17β-estradiol, and testosterone in all skin samples processed through the methanol-based extraction technique presented. Indeed, the results of assay validation demonstrate that EIA can provide reliable measures of all three hormones studied in shark skin extracts. 

As expected, sexually mature males presented significantly higher testosterone levels than females, while immature males and females had similar concentrations. Sex differences in testosterone levels are well-documented in several shark species, with females typically showing lower levels, particularly during the non-breading season [24,35,36], supporting the results of the present study. In females, higher testosterone levels have been related to follicle development [7,36]. In males, testosterone is an important indicator of reproductive activity and sexual maturity [32]. High testosterone values have been related to male maturation processes [35,37], explaining the similar values detected between immature males and females. 

Contrary to expectations, no significant differences were detected between mature and immature males in skin testosterone levels, despite the fact that immature males presented mean concentrations more similar to those of females than mature males. Additionally, no correlation was found between levels of testosterone and GSI in either mature or immature males. Interestingly, testosterone levels in Mediterranean *S.canicula* peak in winter, probably to trigger the mitosis of spermatogonia, inducing spermatogenesis and spermiation in mature males [6]. The lack of differences detected between mature and immature males in the present study could therefore be explained by the fact that all specimens measured were captured in September, during a period when testosterone levels are thought to be lower. Future studies including males sampled within the period of testosterone peak levels would help our understanding of the physiological meaning of shark skin testosterone concentrations. Nevertheless, we must highlight that our sample size in the immature group was considerably smaller, which likely reduced statistical power. A larger sample size could potentially reveal more subtle or group-specific differences. Despite the limited dataset, the observed correlation between the Fulton condition and testosterone levels in immature males suggests that testosterone may play a role in regulating body growth and obtaining fat stores during sexual maturation in the small-spotted catshark. As previously described, testosterone is broadly implicated in the onset of puberty in elasmobranch species [32,38], while achieving a threshold body mass and adequate energy stores may also be beneficial for the initiation of puberty, as described for many mammal species [39,40]. 

Higher progesterone levels were detected in the skin of females compared to both immature and mature males, confirming the hypothesis that sex steroids measured in shark skin could potentially provide relevant biological data. In female oviparous species, such as *S. canicula*, progesterone regulates encapsulation and oviposition [7,25,41]. In males, this hormone has been isolated and measured on several occasions [36,42], and higher circulating levels have been described in females compared to males [24,36]. Progesterone has been described as a precursor of testosterone, and, apparently, this hormone could be involved in male spermatogenesis [43,44]. The functional role of progesterone in males is still a subject of study. Nevertheless, the two-fold greater levels detected in the female group suggest that progesterone measurements in skin hold promise as an approach for addressing physiological questions regarding chondrichthyans. 

Similarly, two-fold greater 17β-estradiol concentrations were detected in female skin samples compared to males in both immature and mature groups. In chondrichthyan females, 17β-estradiol has important reproductive roles, such as the synthesis of vitellogenin [45] or follicle development [41,46]. In males, this hormone also has a relevant function in the gonad’s development and spermatogenesis [24,37,47]. Although comparing 17β-estradiol values between sexes is not a standard practice, few studies suggest that males present lower circulating levels of the hormone compared to females [24,48], as observed in the present study through shark skin. 

Species’ resilience and persistence depend upon their capacity to reproduce and contribute to future generations. The main reproductive activities, such as breeding or mating, are driven by hormone fluctuations. Accordingly, research on reproductive endocrinology is essential for proper management and conservation of threatened species. Common morphological assessments can misjudge sexual maturity, whereas measuring circulating steroid hormones offers a more accurate alternative [37,47]. However, blood collection and other non-lethal methods like endoscopy and ultrasonography involve substantial handling of the animals [49,50]. Skin hormonal analysis could therefore offer a remote sampling alternative to obtain physiological data without the need to chase, capture, and restrain the animal. 

Further research using both male and female chondrichthyans at different maturity stages and phases of the reproductive cycle will provide deep insight into the biological relevance of reproductive hormones in their skin. Moreover, validation across different species and reproductive strategies is required, as diverse breeding and maternal–fetal nutritional strategies in chondrichthyans complicate generalizations about hormonal regulation of reproduction. Skin steroid hormone levels may also be important in providing data on the temporal patterns of reproduction. Therefore, longer time series studies are recommended to identify temporal variations in sex steroid levels and improve our understanding of seasonal and life-stage variations in their reproductive ecology. Additionally, the presented method could benefit important research topics related to chondrichthyan welfare, such as the impact of stress on reproduction and its underlying mechanisms, which remain unclear [51,52]. Elasmobranchs are recognized for their ability to bioaccumulate pollutants, which can have deleterious effects on their reproductive health [53,54]. Therefore, this technique has the potential to be a valuable tool for identifying reproductive dysfunction in this marine species caused by marine pollution.

## 5. Conclusions

The present study demonstrates that sex steroid hormones are deposited in shark skin and that levels can be reliably measured through enzyme immunoassay. Although further validation is required, the observed differences suggest that this technique holds promise as an alternative approach for obtaining reliable biological data in free-ranging sharks. Enhanced knowledge of reproductive information will provide valuable insights into population demographics and viability, which are essential for formulating effective science-based conservation and management strategies.

## Figures and Tables

**Figure 1 animals-15-00762-f001:**
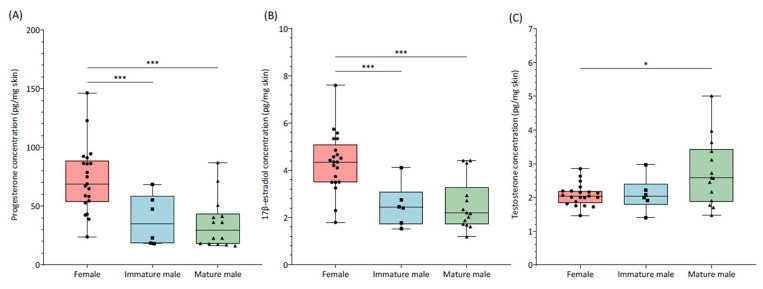
Skin progesterone (**A**), 17β-estradiol (**B**), and testosterone (**C**) levels (pg/mg) in females (N = 21) and immature (N = 6) and mature (N = 14) males of small-spotted catshark (*Scyliorhinus canicula*). The boxplot displays the median value and the interquartile range (25th and 75th percentiles). Whiskers present the highest and lowest values in the dataset. Asterisks represent significant differences between studied groups.

**Figure 2 animals-15-00762-f002:**
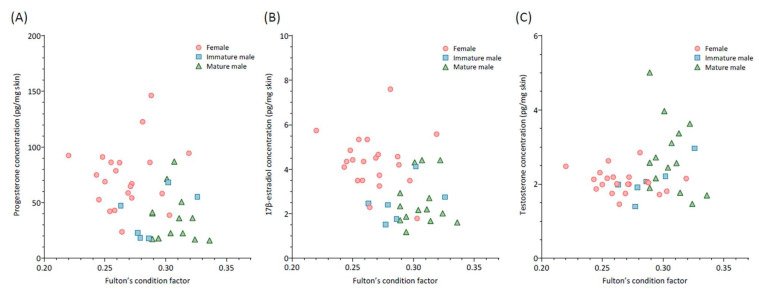
Correlation between skin hormone levels (progesterone (**A**), 17β-estradiol, (**B**) and testosterone (**C**)) and Fulton condition of females (N = 21) and immature (N = 6) and mature (N = 14) males of small-spotted catshark (*Scyliorhinus canicula*).

**Table 1 animals-15-00762-t001:** Biological data (mean ± SD) distributed according to sex and maturity stage (males).

Sex		Total Length (cm)	Body Weight (g)	GSI	*K*	N
Female		39.1 ± 2.2	162.2 ± 36.1	-	0.27 ± 0.02	21
Male	Immature	35.8 ± 2.0	133.2 ± 19.5	1.1 ± 0.5	0.29 ± 0.02	6
	Mature	41.5 ± 2.0	219.9 ± 37.9	3.7 ± 0.6	0.31 ± 0.01	14

GSI, gonadosomatic index; *K*, Fulton’s condition factor; N, number of individuals.

**Table 2 animals-15-00762-t002:** Assay validation of the EIA kit; results show the precision (intra-assay coefficient of variation (CV)), specificity (dilution test), accuracy (spike-and-recovery test), and sensitivity of the measurement of reproductive hormones in skin biopsies of males and females of small-spotted catshark (*Scyliorhinus canicula*). Results are shown as mean (±SD).

Sex	Hormone	Intra-Assay CV (%)	Dilution		Spike-and-Recovery	Sensitivity (ng/mL)
			R^2^ (%)	Mean Error (%)	Mean Recovery (±SD)	
Male	Testosterone *	6.2 (±5.2)	0.99	107.5 (±7.5)	102.2 (±9.7)	0.008
	Progesterone	15.4 (±10.7)	0.99	85.6 (±14.4)	99.2 (±16.3)	0.345
	17β-estradiol	10.2 (±10.7)	0.99	111.5 (±11.5)	87.5 (±19.9)	0.020
Female	Testosterone	4.91 (±2.5)	0.99	109.4 (±9.4)	100.6 (±8.0)	0.012
	Progesterone	10.8 (±5.5)	0.99	111.8 (±11.8)	93.2 (±16.8)	0.196
	17β-estradiol	8.7 (±6.8)	0.99	109.3 (±9.3)	91.5 (±11.9)	0.021

* Results of assay validation for testosterone measurement in male samples were first published in [18].

## Data Availability

Data will be provided upon request.

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
