# Peer review of "A New Research Tool for Use in Sharks and Rays: Relevance of Reproductive Hormone Levels in the Skin of Small-Spotted Catshark (Scyliorhinus canicula)"

_animals, 2025, doi:10.3390/ani15050762_

Round 1
Reviewer 1 Report
Comments and Suggestions for Authors
PLS See attachment
A brief summary:
This MS by Carbajal et al. evaluated whether sex steroid hormones detected in shark skin provide relevant information about their reproductive state. Results of assay validation demonstrated that the commercial enzyme immunoassay used can provide reliable measures of progesterone, 17β-estradiol and testosterone measured in shark skin extracts. After ensuring that reproductive hormones are accurately measured in this sample type, they compared hormone levels between males (mature and immature) and females to evaluate the variation in reproductive hormones relative to sex and maturity stage. We detected significant and biologically relevant differences in skin hormone levels between sexes, suggesting that this technique may be a promising alternative approach for determining reproductive status in free-ranging sharks. By employing this novel technique, they expect to gain a deeper understanding of the reproductive processes of living chondrichthyans, which are essential for formulating effective sci ence-based conservation and management strategies.
Overall, I thought this was a well-executed study in a system with limited previous knowledge of this level on this specific topic. I appreciated their multi-faceted approach (EIA and CV) and the time course involved in this work and think it add significant merit to their work. I do think that their overall conclusions were related to results. I think this is good work, but should undergo major revision before acceptance.
1) Specific comments
1. Introduction
In “Introduction”, the author should introduce the species of small-spotted catshark (Scyliorhinus canicula). Why the author chose this species of shark to do this study? What about the aquaculture output and advantage of small-spotted catshark in Spain? I think this information is very important to show the readers in background, especially the readers from other countries.
2. Materials and Methods
I worry about this part. This study aimed to evaluate whether sex steroid hormones detected in shark skin provide relevant information about their reproductive state. However, in Line 110, “The reproductive stage could not be evaluated in the group of females since their reproductive system had been previously removed and the data was not recorded.” This information is very important, why the author did not obtain this data?
The male was distributed into Immature and Mature stage. I suggest the author should provide the histological section information to show the right developmental stage of the gonad, not only “Immature and Mature stage”.
In addition, I suggest the author supplement the gonad weight of female and the histological section information to show the right developmental stage of the gonad in females.
If this information was lack, I think this is not a complete experiment.
In this study, they want to evaluate whether sex steroid hormones detected in shark skin provide relevant information about their reproductive state. However, they did not make clear the reproductive stage of fish, especially no any information on females. This is very incomprehensible.
Comments on the Quality of English LanguageThe English could be improved to more clearly express the research.
Author Response
Dear reviewer,
Thank you very much for your insightful and constructive feedback on our manuscript. We have carefully considered all of your suggestions and have incorporated the revisions you recommended. The changes are detailed within the attached document.
Thank you again for your valuable contribution to this work.
Best,

Reviewer 2 Report
Comments and Suggestions for Authors
While this is an interesting study and the methods and analyses are appropriate for the tissues collected, the limited sample size and lack of appropriate correlates make it of lesser value. In addition, the authors overreach on their assessment of the value of this type of assessment. With appropriate revisions, this manuscript may be publishable.
In general, the authors claim this method to be less intrusive that the standard blood sampling for sex steroids; however, they provide no evidence to support that supposition. They merely claim that skin biopsies help meet Animal Care and Welfare mandate of the 3Rs. While comparing skin sex hormone levels with estimated maturity state (males only), they provide no direct evidence of reproductive state (e.g., gonad recrudescence). Also, it is unclear why they did not include gonad histology or GSI (Gonadal Somatic Index) as a more appropriate correlate than Fulton Condition Index, since these are the direct gonadal correlates used to determine reproductive state. While a well provisioned individual with high Fulton condition index might exhibit a more robust reproductive state there is no direct correlation to gonadal development relative to skin sex hormone levels. Additional, circulating blood sex hormone levels are typically used and well correlated method for assessing reproductive state. This tried-and-true methodology may be argued to be even less intrusive than skin biopsy methods in the hands of a skilled technician, where blood samples can be quickly and safely acquired from selachians held in tonic immobility (less than 2 min). Which also raises the question as to why authors did not sample blood plasma (standard practice) for correlations with skin sex hormone levels to demonstrate whether they are correlated with gonadal state? Interestingly, skin biopsy methodology may actually be far more appropriate for some batoids, especially small species, that are particularly difficult to non-lethally blood sample. However, without some direct correlation to blood plasma hormone levels, this too remains subjective. Conclusions regarding efficacy are also greatly limited by the disproportionate sample sizes among sexes and estimated maturity state. While I can appreciate the challenges of acquiring sufficient samples, I believe the authors overstate the value of their conclusions, despite the mention of this challenge.
Additionally, circulating sex hormones in blood may be a far more appropriate metric for assessing reproductive state, since it often reflects more immediate responses from gonads via the HPG axis. The authors provide no evidence that acquiring skin biopsies is any less traumatic. Unfortunately, their methods were trialed on deceased animals, preventing any assessment of 1) reduction in pain, 2) energetic costs and assurances of healing, 3) lower rates of infection in comparison with blood sampling. Therefore, the arguments that this method is better than the standard sex hormone sampling practices remain highly subjective. With all that said, I do find the results quite interesting and think if the authors “tone down” their perceived ethics value and conclusions, this may be a valuable contribution.
Lastly, I was disappointed to see that there was little discussion as to why skin samples would contain proportionally high sex hormone levels. While the authors casually mention that sex hormones in vertebrates also serve other non-reproductive functions for tissues and behavior, they provide no explanation as to why these hormones are found at proportional levels across sexes and maturity states. There is a growing body of literature describing this in other taxa (e.g., marine mammals), where epidermal mucus swabs and breath exhalation have been used as sex hormone – reproductive state correlates.
Because of the limited sample size, lack of correlation between actual reproductive state, I would highly recommend authors greatly reduce Introduction and Conclusions, reducing this manuscript to the equivalence of a note or brief communication.
Author Response

(The authors gave the same response as above.)

Round 2
Reviewer 1 Report
Comments and Suggestions for Authors
I think the author has provided point-to-point modifications, and I agree to publish in the current format.
Comments on the Quality of English LanguageThe English could be improved to more clearly express the research.
Author Response
Thank you for your thorough review and for your acceptance of the revisions. We appreciate your contribution to improving the manuscript
Reviewer 2 Report
Comments and Suggestions for Authors
Overall, there are many improvements; however, there is still some confusion in regards to terms, writing and conclusions.
Line 33: It is unclear what is meant by “remotely’? Do you mean sampling animals in the field for sample and release, or do you mean using some type of device to obtain a skin sample from an animal completely unrestrained? Either way, this term needs to be defined or the reader will not understand what is meant here. It is particularly problematic here in abstract.
Line 35: this should specify “skin biopsies”. Skin alone is too vague. You’re talking about a field base sampling tool like a biopsy punch, not just taking big chunks of skin.
Lines 53-55: I think this sentence needs to be written. Essentially, the point of the sentence is the value of finding non-lethal, minimally invasive methods for assessing physiological processes and states of wild sharks and rays. This sentence does not clearly state that.
Line 65: do you mean “non-lethal’? Not sure what “non-destructive” means here. Some would argue that taking a skin/muscle biopsy is more destructive than a caudal vein puncture.
Line 68: Again, no idea what the term “remotely” means here.
Line 71: It’s not until this paragraph that the authors start to articulate their real point of sampling, which is really about skin biopsy sampling from uncaptured wild individuals. If this is the main point then it needs to be made clear in the beginning of the Introduction when authors are talking about how this method matters in terms of the 3Rs. Sounds like the goal to not require capture or handling at all. If that is the point, then that needs to be clearer above.
Line: 159-168: I don’t mean to bring up something new, but it seems like the goal is to provide a correlation between hormone level with 1) size at maturity and 2) seasonal reproductive state. If that is the case, it can only be done for males, then why not just do a logistic regression with reproductive state (reproductively functional, non-functional) and then androgen conc. and size as independent variables? Not that this is required, but it would simplify this analysis.
Line: 235: should be specific here – skin biopsies, not just skin.
Lines: 237-246: much better and clearer rationale.
Line 314: remote sampling unclear and undefined.
Author Response
Dear Reviewer,
Thank you for your helpful comments. We have addressed each point and have attached a document with our point-by-point responses. We appreciate your time and consideration.
